

# Simulating the effects of temperature and precipitation change on vegetation composition in Arctic tundra ecosystems

H. van der Kolk[1], M. M. P. D. Heijmans[1], J. van Huissteden[2], J. W. M. Pullens[1,3,4], F. Berendse[1]

[1]Plant Ecology and Nature Conservation Group, Wageningen University, Wageningen, The Netherlands
[2]Earth and Climate Cluster, VU University Amsterdam, Amsterdam, The Netherlands.
[3]Sustainable Agro-Ecosystems and Bioresources Department, Research and Innovation Centre – Fondazione Edmund Mach, San Michele all' Adige, Italy
[4]Hydromet, Department of Civil and Environmental Engineering and Environmental Research Institute, University College Cork, Cork, Ireland

*Correspondence to*: M. M. P. D. Heijmans (monique.heijmans@wur.nl)

**Abstract.** Over the past decades, vegetation has changed significantly along with climatic changes in the Arctic. Deciduous shrub cover is often assumed to expand in tundra landscapes, but more frequent abrupt permafrost thaw resulting in formation of thaw ponds could lead to vegetation shifts towards graminoid dominated wetland. Which mechanisms drive vegetation changes in the tundra ecosystem is still not sufficiently clear. In this study, the dynamic tundra vegetation model
NUCOM-tundra was used to evaluate the consequences of climate change scenarios of warming and increasing precipitation for future tundra vegetation change, and to identify the mechanisms that drive these changes. The model includes three plant functional types (moss, graminoids and shrubs), carbon and nitrogen cycling, water and permafrost dynamics and a simple thaw pond module. Climate scenario simulations were performed for sixteen combinations of temperature and precipitation increases in five vegetation types representing a gradient from dry shrub dominated, to moist mixed and wet graminoid
dominated sites. Vegetation composition dynamics in currently mixed vegetation sites was dependent on both temperature and precipitation changes, with warming favouring shrub dominance and increased precipitation favouring graminoid abundance. Climate change simulations based on greenhouse gas emission scenarios in which temperature and precipitation increases were combined showed initial increases in graminoid abundance followed by shrub expansion with further climate change. The simulations suggest that the shrubs are better light competitors, but their growth can be limited by very wet soil
conditions and low nutrient supply. Graminoids have the advantage of being able to grow in a wide range of soil moisture conditions and having access to nutrients in deeper soil layers. Abrupt permafrost thaw initiating thaw pond formation led to complete domination of graminoids. However, due to increased drainage, shrubs could profit from such changes in adjacent areas. Both climate and thaw pond formation simulations suggest that a wetter tundra can be responsible for local shrub decline instead of shrub expansion.

*Key words*: Ecosystem model; Climate change; NUCOM-tundra; Permafrost thaw; Shrub expansion; Thermokarst; Tundra vegetation.



## 1 Introduction

Tundra ecosystems in the Arctic are shaped by strong interactions between biological, hydrological and climatological factors (Hinzman et al., 2005). An important feature of the Arctic ecosystem is permafrost, which is the soil that is persistently frozen for at least two years. For the near future, global climate models project a further increase in temperature

and precipitation with pronounced changes in the Arctic region (Johannessen et al., 2004; Vavrus et al., 2012; IPCC, 2014). Rising temperatures potentially result in increased seasonal thawing of the permafrost and thus in an increased thickness of the active layer (the soil layer that thaws during the growing season) and drawback of near surface permafrost (Anisimov et al., 1997; Lawrence and Slater, 2005; Zhang et al., 2005). Given the large soil organic carbon stocks in permafrost (Zimov et al., 2006a; Tarnocai et al., 2009), there is a major concern that these stocks get decomposed and subsequently released as

$CO_2$ and $CH_4$ to the atmosphere (Dutta et al., 2006; McGuire et al., 2009). An increase in greenhouse gas release, including $CO_2$ and $CH_4$ (Christensen et al., 2004; Walter et al., 2006; Nauta et al., 2015; Schuur et al., 2015), from thawing permafrost could stimulate further warming and result in a positive feedback loop between temperature increase and greenhouse gas emission (Zimov et al., 2006b; Schuur et al., 2008; MacDougall et al., 2012; van Huissteden and Dolman, 2012; Schuur et al., 2015).

Climate change in the Arctic region affects tundra vegetation composition. The northernmost tundra is dominated by mosses and lichens due to the extreme low summer temperatures. Southwards, with increasing summer temperatures, graminoids and dwarf shrubs increase in abundance (Walker et al., 2005). Climate change influences the tundra vegetation in multiple ways. Warming experiments in tundra ecosystems showed an increase of graminoids and deciduous shrubs in response to raised temperatures, while mosses and lichens and the overall species diversity decreased (Walker et al., 2006). Shrubs have

been observed to expand with ongoing temperature increase presumably due to the increased availability of nutrients in the warmer soil (Tape et al., 2006; Myers-Smith et al., 2011). Several tree and shrub species, including dwarf birches (*Betula glandulosa* and *Betula nana*), willows (*Salix* spp.), juniper (*Juniperus nana*) and green alder (*Alnus viridis*), have expanded and increased in abundance in the Arctic as a response to climatic warming (Sturm et al., 2001; Tape et al., 2006; Hallinger et al., 2010; Elmendorf et al., 2012). However, besides climate, other factors as herbivory, soil moisture and soil nutrient

availability affect shrub growth as well and it is therefore complex to predict the expansion of shrubs in the Arctic region (Myers-Smith et al., 2011, 2015). Increased abundance of shrubs might have important consequences for permafrost feedbacks. For example, an increase of low shrubs might slow down permafrost thaw as a result of the shadow they cast on the soil (Blok et al., 2010). However, tall shrubs may increase atmospheric heating and permafrost thawing due to their lower albedo (Bonfils et al., 2012)

Another mechanism by which climate change affects tundra vegetation is abrupt thaw resulting in local collapse of the permafrost. Thawing of underground ice masses results in a collapse of the ground, by which water filled thermokarst ponds are formed. These ponds are first colonized by sedges and later by mosses (Jorgenson et al., 2006). Due to thaw pond



formation, changes in northern permafrost landscapes from dry birch forests or shrub dominated vegetation towards ponds or wetlands dominated by graminoids may occur with climatic change in both discontinuous and continuous permafrost regions (Jorgenson et al., 2001; Turetsky et al., 2002; Christensen et al., 2004; Jorgenson et al., 2006; Nauta et al., 2015). In the continuous permafrost region, permafrost collapse can change the vegetation from shrub dominated towards a wet graminoid dominated stage within less than one decade. The thermokarst ponds might stimulate further soil collapse and consequently drastically alter hydrological and soil processes, also in adjacent areas (Osterkamp et al., 2009; Nauta et al., 2015; Schuur et al., 2015).

The mechanisms that drive the observed tundra vegetation composition changes, especially shrub expansion, and their consequences are not yet well understood. Increased air temperatures lead to higher soil temperatures, especially in the shallow layers, and may thus promote microbial activity, thereby increasing nutrient availability in the soil. In tundra vegetation, shrubs are hypothesised to be best able to respond to such increased nutrient availability (Tape et al., 2006). It is, however, not known to what extent this and other shrub growth factors, including precipitation, growing season length and disturbances (Myers-Smith et al., 2011), will affect the competition between different plant functional types in the Arctic. As shrub expansion has important implications for land surface albedo and consequently climate feedbacks (Chapin et al., 2005; Pearson et al., 2013), it is crucial that we understand and are able to predict further vegetation change in the tundra.

One approach to better understand the interactions between plants in tundra landscapes is to use a dynamic vegetation model to analyse developments in vegetation composition in response to climatic changes. Tundra vegetation models that aim to predict the impacts of climate change on vegetation-substrate interactions should at least include the most important plant functional types, competition and permafrost feedbacks. Although several tundra vegetation models exist, these models do not take into account hydrological feedbacks, the formation of thaw ponds and vegetation-permafrost feedbacks or do not include mosses as separate plant functional type (e.g. Epstein et al., 2000; Wolf et al., 2008; Euskirchen et al. 2009). Therefore, in this study, the effects of climate change, including both temperature and precipitation change, on plant competition and vegetation composition were studied by developing a new model named NUCOM-tundra based on earlier NUCOM (NUtrient and COMpetition) models for other ecosystems (Berendse, 1994a, 1994b; van Oene et al., 1999; Heijmans et al., 2008, 2013). The new tundra vegetation model includes mosses, graminoids, dwarf shrubs, hydrological and soil processes, and permafrost dynamics. In this study, we simulated tundra vegetation changes under different climate change scenarios in order to (1) analyse the effects of future temperature and precipitation scenarios on tundra vegetation composition, (2) identify important mechanisms that drive these vegetation changes, and (3) explore the impacts of thaw pond formation due to local permafrost collapse.

## 2 Methods

### 2.1 Brief model description



NUCOM-tundra has been developed to simulate long-term dynamics of vegetation composition in tundra landscapes for analysis of vegetation-permafrost-carbon feedbacks in relation to climate change and includes nitrogen and carbon cycling, permafrost and water dynamics (Fig. 1). An extensive model description and all equations are provided in Supplement Section S1.

NUCOM-tundra simulates the dynamics of three Plant Functional Types (PFTs), moss, graminoids (e.g. *Eriophorum vaginatum*) and deciduous dwarf shrubs (e.g. *Betula nana*), on a local scale at the decadal timescale using a daily time step. NUCOM-tundra represents tundra landscapes, which are an alternation of shrub dominated, graminoid dominated and mixed vegetation types. The model is based on an area of 1 $m^2$. The biomass and nitrogen content of the vascular plant PFTs is separated into fine roots, woody plant parts (for shrubs) or rhizomes (for graminoids), and leaves. Mosses form a layer on

top of the soil surface, with a thickness up to 4.5 cm. The soil profile is divided into an organic top soil layer with a variable height and 10 deeper mineral soil layers each with a thickness of 10 cm. The fine roots of vascular plants are distributed throughout the active soil profile, with graminoids rooting deeper in comparison to dwarf shrubs (Shaver and Cutler, 1979; Nadelhoffer et al., 1996; Iversen et al., 2015). The thickness of the active layer depends on the soil temperature profile (Supplement Section S1). The thickness of the organic layer, which consists solely of moss, leaf, stem and root litter,

generally increases over time, depending on the balance between litter input and litter decomposition. The mineral soil layers contain initial soil organic carbon and nitrogen pools as well. During the simulations, only fine root litter is added to the mineral soil layers and decomposed there. The other litter types become part of the soil organic layer. Decomposition rates depend on soil temperature and differ among PFTs and plant organs (leaves, stems and fine roots).

Plant growth (net primary production) is determined by temperature, light availability, nutrient availability and moisture

conditions (Supplement Section S1). In the model, there are temperature thresholds for plant growth thus excluding growth during the winter season. Graminoids and dwarf shrubs compete for the incoming light based on their leaf area. It is assumed that graminoids and dwarf shrubs are equally tall. The moisture content in the upper 10 cm of the soil can strongly reduce PFT growth as both graminoids and dwarf shrubs have an optimum growth only in part of the range of possible soil moisture conditions. Dwarf shrubs prefer drier conditions, and cannot grow if the soil is completely water saturated. The graminoids

in the model do not grow well under dry conditions, but can grow on water-saturated soils.

Mosses acquire nitrogen by nitrogen fixation from the atmosphere and can absorb available nitrogen from the upper cm of the soil profile. Vascular plant nitrogen uptake is determined by the fine root length of both graminoids and dwarf shrubs present in each layer and the amount of nutrients available. At the start of the growing season, when the air temperature is above the threshold for growth but the soil is still mostly frozen, the plants can use nitrogen from an internal storage, which

is filled by nitrogen reallocated from senescing leaves and roots. Dying plant material enters the soil organic carbon and nitrogen litter pools. Soil organic carbon is lost by microbial decomposition, whereas the mineralised nitrogen from soil



organic nitrogen becomes available for plant uptake. Subsequently, part of this pool of available nitrogen can be lost by denitrification under high soil moisture conditions (Supplement Section S1).

A simple hydrological module is included in NUCOM-tundra, which simulates the volumetric water content of the organic and mineral soil layers (Supplement Section S1). Water from snowmelt, rainfall and inflow from a neighbouring vegetation type (Fig. 2) fills up the pore space in the soil layers. Evapotranspiration, surface runoff, and lateral drainage out of the moss and organic layer lower the water content. The hydrological processes follow a seasonal pattern. A snow layer accumulates during the winter season, and snowmelt occurs with positive air temperatures at the start of the growing season. During this period, the shallow active layer becomes water saturated and the excess of water runs off over the soil surface. During the growing season, evapotranspiration generally exceeds precipitation, resulting in gradual drying of the soil.

## 2.2 Parameter values and model input

Parameter values for plant properties such as root characteristics, mortality, reallocation and decomposition have been derived from literature or have been calibrated using field data collected at the Chokurdakh Scientific Tundra Station, located 70° 49′ N, 147° 29′ E, altitude 10m (Supplement Section S2). The Chokurdakh Scientific Tundra Station is located in the lowlands of the Indigirka river, north-eastern Siberia, which is in the continuous permafrost zone and the Low Arctic climate zone. The vegetation is classified as tussock-sedge dwarf-shrub moss tundra (vegetation type G4 on Circumpolar Arctic Vegetation Map; Walker et al., 2005).  The parameter values are provided in Appendix A.

NUCOM-tundra requires daily temperature and precipitation data as input for the model simulations. Weather data from Chokurdakh Weather Station (WMO station 21946, located 70˚ 62‘ N, 147˚ 88‘ E; altitude 44.0m) for the years 1954 to 1994 obtained through the KNMI climate explorer tool (Klein Tank et al., 2002, http://climexp.knmi.nl) were input to the model. Initial values of the model simulations are provided in Supplement Section S3. The biomass start values were based on field measurements at the Chokurdakh Tundra Station.

### 2.3 Vegetation types

We defined five initial vegetation types representing a gradient from relatively dry to wet sites for the climate change simulations in NUCOM-tundra. These initial vegetation types represent five landscape positions, ranging from relatively well-drained shrub patches to waterlogged graminoid-dominated wetland (Fig. 2). The first vegetation type represents dry shrub-dominated vegetation. Vegetation types 2, 3 and 4 represent moist mixed vegetation from relatively dry to wet sites. Vegetation type 5 represents a wet graminoid-dominated vegetation downslope which receives water from the neighbouring cell. In vegetation type 1, only outflow of both surface water and water in the organic layer occur, whereas moist sites are characterized by having both water inflow and water outflow. The water flow into a downslope landscape position is the actual water outflow from the adjacent upslope located simulation cell.





## 2.4 Comparison with field data

The performance of several model parts was evaluated by comparing outputs with data from the Chokurdakh Tundra Station and Weather Station. The simulated timing of snow accumulation and snowmelt was compared with snow height data from the Chokurdakh Weather Station between 1944 and 2008. Furthermore, simulated PFT total biomass, biomass partitioning among leaves, woody parts, rhizomes and fine roots and vertical root distribution were compared with field data. Field biomass collections were done for both graminoids and dwarf shrubs in graminoid dominated, mixed and dwarf shrub dominated vegetation sites at the tundra field site in north-eastern Siberia (Wang et al., 2016). Roots were collected over different depths, enabling to compare the field data with simulated root distribution over depth. These field data were compared with simulations that were run with climate input from the Chokurdakh Weather Station. A detailed description of these comparisons of simulated and field data is provided in Supplement Section S4.

## 2.5 Climate scenario simulations

For all vegetation types a period of 40 years was simulated to initialize the model, using precipitation and temperature data from the Chokurdakh weather station from 1 August 1954 till 31 July 1994 as input. This period excludes the most pronounced warming that took place over the last decades. Annual precipitation was, however, variable with both increasing and declining trends in the 40-year time period (Supplement Section S2). After the initialisation phase, 16 different climate change scenarios were run for all five vegetation types for the period 1 August 1994 – 31 July 2074. Inspired by the RCP emission scenarios over the 21[st] century, we combined four temperature increases with four precipitation increases to simulate climatic changes over this 80-year period. The combinations include temperature increases of 0, 2.5, 5 or 8 $^{\circ}$C and precipitation increases of 0, 15, 30 or 45mm per 100 years. In the Arctic region, an increase of 2.5 $^{\circ}$C and 15mm precipitation is expected under emission scenario RCP2.6 and an increase of 8 $^{\circ}$C and 45 mm precipitation are expected over the 21[st] century under emission scenario RCP8.5 (IPCC, 2014). A scenario with 5$^{\circ}$C and 30 mm precipitation change is regarded here as an 'Intermediate' scenario.

The weather data series over the period 1 August 1954 – 31 July 1994 from the Chokurdakh weather station was used twice to create a baseline (no climate change) for the climate scenario simulations between 1994 and 2074. For all climate change scenarios, temperature and precipitation were gradually increased over this 80 year simulation period. For precipitation, only the intensity of rainfall was increased and not the number of days at which rainfall occurred.

To evaluate the effects of climate change on the vegetation composition, we compared the biomass of moss, graminoids and dwarf shrubs on 31 July, averaged over the last 10 years of the simulation (corresponding with 2065-2074). Furthermore, several factors that influenced plant growth and competition were evaluated using the simulations for moist mixed vegetation sites (vegetation type 3) in which all three PFTs co-occur. The following factors were evaluated for the last 10 years of the simulations:



1) *Growing season length*: The average number of days per year at which the temperature threshold for vascular plant growth was exceeded;

2) *Soil moisture*: Average soil moisture content in the upper 10cm of the soil profile during the growing season;

3) *Light competition*: The average percentage of light intercepted by graminoids and shrubs during the growing season;

4) *Moisture conditions*: Percentage of time in the growing season with optimal soil moisture content for growth of graminoid and dwarf shrub;

5) *Nutrient limitation*:  Limitation of growth of graminoid and dwarf shrub due to insufficient available nutrients in the growing season, expressed as percentage of potential growth realised.

**2.6 Thaw pond simulations**

Thaw pond initiation was simulated for dry and moist vegetation types (vegetation types 1-4) under three different climate change scenarios: no change, RCP2.6 and RCP8.5. Thaw pond collapse was simulated by imposing a sudden alternation of water flows. Upon a collapse event, water inflow, including surface runoff and lateral drainage through the soil organic layer, doubled whereas water outflow stopped. To evaluate the effects of thaw pond collapse on downslope vegetation sites, simulations were performed in series that included all five vegetation types (Fig. 3). Thaw pond formation was initiated at a fixed time step halfway the simulation (corresponding to 31 July 2034). The change in vegetation composition after collapse was evaluated by determining the abundance of graminoids (in %) in the vascular plant community biomass in the last 10 summers of the simulation (31 July in the years 2065-2074), when the vegetation is at its peak biomass.

**3 Results**

**3.1 Comparison with field data**

Simulated snow height, total PFT biomass, biomass partitioning and vertical root distribution were compared with observations from Chokurdakh Scientific Tundra Station or Weather Station (detailed description in Supplement Section S4). Simulated timing of accumulation and melt of the snow layer between 1944 and 2008 was well in agreement with observations of snow depth at the Chokurdakh Weather Station. Furthermore, NUCOM-tundra simulated total biomass for graminoids and dwarf shrubs within the range of biomass values obtained for shrub, mixed and graminoid vegetation types at the field site. Simulated biomass partitioning over leaves, stems and fine roots was comparable to the partitioning observed in the field with high total biomass, although the simulated biomass partitioning was rather constant and did not change with increasing total biomass, as observed in the field. Fine root vertical distribution patterns were similar between observed and simulated data as NUCOM-tundra took into account the different rooting patterns of graminoids and dwarf shrubs, and variations in active layer thickness.





## 3.2 Climate scenario simulations: Vegetation changes

In the No climate change scenario, the total vegetation biomass was rather stable throughout the simulation in all vegetation types (Fig. 4A for vegetation type 3, data not shown for other vegetation types). Gradual biomass increases between 1994 and 2074 were simulated under the three climate change scenarios based on the RCP scenarios (Fig. 4B, 4C, 4D).

5 Considering all combinations of temperature and precipitation scenarios in all vegetation types, shows that the simulated total biomass of the vegetation responded strongly to increased temperature (Fig. 5). The average total (aboveground and belowground) summer biomass between 2065 and 2074 ranged from 1544 g m$^{-2}$ to 2428 g m$^{-2}$ with no temperature change, whereas total biomass ranged between 3702 g m$^{-2}$ to 4483 g m$^{-2}$ with a gradual increase of 8 $^{o}$C over the 21$^{st}$ century. In contrast, precipitation change did not affect the total vegetation biomass.

10 The vegetation composition in relatively well-drained sites dominated by shrub vegetation (vegetation type 1) and wet graminoid-dominated vegetation (vegetation type 5) did not change under any of the climate change scenarios (Fig. 5). In mixed vegetation (vegetation type 3), graminoids slightly increased in abundance with the RCP2.6 scenario (2.5 $^{o}$C temperature and 15mm precipitation year$^{-1}$ increase over the 21$^{st}$ century) (Fig. 4B; Fig. 6). However, under scenario RCP8.5 (8 $^{o}$C temperature and 45 mm precipitation year$^{-1}$ increase over the 21$^{st}$ century), shrubs became more dominant by 15 2065-2074 after an initial increase in graminoid abundance (Fig. 4D). In other climate change scenarios, the magnitude of both precipitation and temperature change determined changes in vegetation composition in currently mixed vegetation sites (Fig. 6). The proportion of graminoids in the vascular plant community increased in simulations with more pronounced precipitation changes and lower temperature changes (Fig. 4E; Fig. 6). In contrast, graminoids were outcompeted by shrubs by the end of simulations of initially mixed vegetation sites under a scenario with 8$^{0}$C temperature increase but no 20 precipitation change over the 21st century (Fig. 4F).

Simulated moss biomass was in general lowest in wet graminoid dominated vegetation sites (Fig. 5), and showed little variation among the climate scenario simulations (Fig. 4; Fig. 5).

## 3.3 Nutrient availability, light competition and moisture conditions

In comparison to the scenario for no climate change, the growing season was on average 39 days longer between 1 August 25 1964 and 31 July 1974 in simulations with an 8 $^{0}$C temperature increase scenario (Table 1). During the same time period, the averaged soil moisture at mixed vegetation sites was higher in simulations with more pronounced precipitation change and lower temperature changes (Table 1). Consequently, soil moisture conditions became more favourable for graminoids under these scenarios, whereas moisture conditions became more favourable for shrubs with large changes in temperature and small changes in precipitation. For the RCP2.6, intermediate and RCP8.5 scenario, moisture conditions became less 30 favourable for graminoid growth and more favourable for shrub growth in comparison to the scenario for no climate change. In general, shrubs intercepted more light in comparison to graminoids (Table 1). Both graminoids and shrubs intercepted more light when moisture conditions became more favourable. The effect of temperature changes on light interception





differed between graminoids and shrubs. Shrubs intercepted more light when temperature changes were more pronounced. Light interception in graminoids, however, decreased under scenarios with 8 $^{0}$C temperature increase and less pronounced precipitation increase over the 21$^{st}$ century. Nutrients limited shrub growth under all climate scenarios (Table 1). For both graminoids and shrubs, nutrients became less limiting under less suitable growth conditions.

## 3.4 Thaw pond simulations

All thermokarst events (initiated in 2034) led to the complete domination of graminoids over shrubs within 15 years after initiation of thaw pond development (Fig. 7; Fig. 8). The vegetation composition on collapsed sites became similar to the composition on wet graminoid dominated sites (vegetation type 5). After abrupt permafrost thaw, bryophytes took advantage of the decreased vascular plant leaf area, but stabilized later at a biomass that was equal to the moss biomass in non-collapsed wet graminoid dominated vegetation sites (Fig. 8). When a collapse occurred in vegetation type 1, 2 or 3, the collapse enhanced shrub expansion in the next grid cell. The thermokarst pond acted as a water sink, due to which water flows into the next grid cell were halted. Consequently, the grid cell adjacent to the thermokarst pond became drier which favoured shrub growth (Fig. 7). Similar patterns for the thaw pond simulations were observed for No change, RCP2.6 and RCP8.5 climate scenarios.

## 4. Discussion

The effects of climate change on tundra vegetation are complex since changes in hydrology, active layer depths, nutrient availability and growing season length interact with each other (Serreze et al., 2000; Hinzman et al., 2005). Climate warming might alter vegetation composition by increasing nutrient availability due to faster mineralization rates and increasing active layer depths (Mack et al., 2004). It remains, however, unclear which factors are most decisive with respect to the observed expansion of shrubs at the expense of graminoids (Myers-Smith et al., 2011). NUCOM-tundra was developed to explore future Arctic tundra vegetation composition changes in response to climate scenarios and to assess which mechanisms are responsible for these changes.

### 4.1 Vegetation Composition Changes

Our climate scenario simulations suggest a significant increase in biomass with continuing climate change, and eventually increased shrub abundance under scenarios with strong climate change, such as the RCP8.5 emission scenario. For the modest emission scenarios (RCP2.6, +2.5 $^{o}$C and +15mm precipitation over the 21$^{st}$ century), particularly graminoid biomass increased. These simulations are in line with biomass increases in both shrubs and graminoids that have been observed during the past decades in Arctic tundra landscapes as a response to temperature increase (Dormann and Woodin, 2002; Walker et al., 2006; Hudson and Henry, 2009). In addition, shrub expansion, as predicted by the model for strong climate




change scenarios, has been observed in many places in the Arctic (Sturm et al., 2001; Tape et al., 2006; Myers-Smith et al., 2011).

As nutrient availability limits plant growth in tundra landscapes (Shaver et al., 2001), a positive effect of warming on nutrient availability is a likely explanation for biomass increase observed in tundra vegetation (Hudson and Henry, 2009).

Climate warming might influence nutrient availability positively by lengthening of the growing season, active layer deepening and increased microbial activity. In our simulations, nutrients were especially limiting for the shrubs, due to their shallow rooting systems. Compared to shrubs, graminoids root relatively deep. As a consequence, active layer deepening is expected to favour especially graminoids. It is, however, unclear how plant root morphology responds to climate warming. An experimental warming study in dry tundra demonstrated that plants do not necessarily root deeper in response to warmer

temperatures, but instead may concentrate their main root biomass in the organic layer where most nutrient mineralization takes place (Björk et al., 2007). Nevertheless, it is likely that growing season lengthening and increased microbial mineralization of soil organic matter improve growing conditions for both shrubs and graminoids, as also graminoids have been shown to respond strongly upon fertilization (e.g. Jonasson, 1992).

The effectiveness of shrubs to deal with increased nutrient levels has been proposed as an explanation that shrubs can expand

in a warmer climate (Shaver et al., 2001; Tape et al., 2006). *Betula nana* responds to higher nutrient availability by increasing its biomass, which is mainly due to increased secondary stem growth (Shaver et al., 2001). However, *Betula nana* is also known to respond to increased temperatures and fertilization by growing taller and by producing more shoots and tillers, thereby increasing its ability to compete for light with other species (Chapin and Shaver, 1996; Hobbie et al., 1999; Bret-Harte et al., 2001; Shaver et al., 2001). Although tiller production is not explicitly included in NUCOM-tundra, shrubs

clearly had an advantage in the competition for light as they have a higher specific leaf area and higher light extinction coefficient than the graminoids. With warming-induced increases in aboveground biomass for both graminoids and shrubs, the competition for light becomes more important.

The climate simulations in this study show that shifts in vegetation composition are not only dependent on temperature change, but are strongly affected by precipitation changes as well. Simulated soil moisture contents decreased with higher

temperature and lower precipitation scenarios. Evapotranspiration is an important hydrological process determining soil moisture during the growing season. Throughout the growing season, the top soil layer dries out as evapotranspiration exceeds precipitation during this period (Supplement Section S2). Higher summer temperatures increase potential evapotranspiration, and thus lead to drier soils, if precipitation remains unchanged. Consequently, the area of dry vegetation sites characterized by their dense dwarf shrub coverage might increase. A similar misbalance of temperature and

precipitation change between 1950 and 2002 has been proposed by Riordan et al. (2006) as one of the possible causes for drying of thermokarst ponds in Alaska. A second mechanism for tundra drying with higher temperatures is increased water drainage enabled by gradual deepening of the active layer or permafrost degradation. The latter mechanism is especially





important in the discontinuous permafrost zone, where climate change may cause loss of permafrost at thermokarst sites and subsequently lead to increased drainage to adjacent areas (Yoshikawa and Hinzman, 2003).

Predicted increases in total biomass and expansion of shrubs are not unique for NUCOM-tundra, but have also been simulated by other Arctic vegetation models (e.g. Epstein et al., 2000; Euskirchen et al., 2009; Yu et al., 2011). Our results,

however, highlight the importance of changes in annual precipitation, as they have a large influence on the water table and thus on the tundra vegetation composition. Precipitation, especially in summer, compensates for evapotranspiration water losses, thereby having a direct positive influence on soil moisture and water table position. Climate models predict considerable precipitation increases for the Arctic region (IPCC, 2014) as the retreat of sea ice in the Arctic Ocean results in strongly increased evaporation and precipitation (Bintanja and Selten 2014). In agreement with these predictions, significant

increases in precipitation over the last decades have been demonstrated for several Arctic weather stations (Hinzman et al., 2005; Urban et al., 2014). Changes in precipitation are, however, less consistent in comparison to climate warming (Urban et al., 2014). In Chokurdakh, the location used for the model simulations, precipitation has not increased along with temperature change between 1981 and 2010 (Temperature +0.0565 $^\circ$C year$^{-1}$; Precipitation -1.127 mm year$^{-1}$) (climexp.knmi.nl/atlas; Li et al., 2015). Continuation of the climate trends observed in Chokurdakh between 1981 and 2010

would lead to a scenario that most resembles a temperature increase of 5$^\circ$C and no precipitation change by the end of this century. This scenario would imply that currently mixed vegetation will shift towards shrub dominated vegetation in the near future due to tundra drying.

## 4.2 Thaw Pond Formation

Formation and expansion of shrub vegetation might be set back by the degradation and collapse of ice wedges in the

permafrost. Even in dry vegetation type simulations, the initiation of thermokarst led, as a result of water accumulation in thaw ponds, to complete replacement of shrubs by graminoids. The simple thaw pond module in NUCOM-tundra simulated a peak of moss biomass shortly after ground collapse and before significant colonization of graminoids. After the collapse, the shrubs were dying because of the water-saturated soil, while the graminoids had not yet colonized the new thaw pond, and the mosses took advantage of the increased light availability. At the Siberian field site we observe that mosses mostly

drown as well and the first colonizers of thaw ponds are mostly graminoids, for example *Eriophorum angustifolium* (Jorgenson et al., 2006), although also mosses can be first colonizers. NUCOM-tundra does not (yet) take surface water depth into account for the calculation of plant and bryophyte growth, due to which drowning of mosses could not be simulated. The model is also not able to simulate further vegetation succession in thaw ponds. Most likely, after graminoid and moss establishment, an organic layer starts to accumulate, shrubs recolonize the collapsed sites, after which permafrost

aggradation can take place. Water runoff out of collapsed sites is expected to increase when the organic layer accumulates, probably aided by newly developing ice structures which lift the surface.



Simulated thermokarst events favoured shrub growth in adjacent grid cells due to increased drainage. Although graminoids become dominant in thaw ponds, ice wedge collapses may provide opportunities for shrub growth on the pond margins due to small scale changes in water flows. Such a mechanism was observed in Alaska, where shrubs became dominant around thermokarst spots as a result of increased drainage from shrub-dominated patches to the new ponds (Osterkamp et al., 2009).

These results further reveal that tundra drying is an important mechanism for shrub expansion in the Arctic. Thermokarst ponds, however, may also act as a heat source and thereby stimulate further permafrost collapse (Li et al., 2014). Overall, permafrost degradation should be recognized as an important potentially graminoid stimulating factor when studying climate change in Arctic landscapes (Jorgenson et al., 2006). Rapid expansion of shrubs might be partly compensated by the formation of graminoid dominated thaw ponds, particularly in poorly drained lowland tundra.

The inclusion of thaw pond formation is a new but not yet fully developed element of NUCOM-tundra. There are models that simulate tundra vegetation change (ArcVeg (Epstein et al., 2000); TEM-DVM (Euskirchen et al., 2009); LPJ-Guess (Wolf et al., 2008; Zhang et al., 2013)), and models that simulate thaw lake cycle dynamics (van Huissteden et al., 2011). Taking into account the formation of thaw ponds and the subsequent vegetation succession is, however, essential for a full analysis of climate change effects on tundra ecosystem (Van Huissteden & Dolman, 2012). This model feature should

therefore receive special attention with further model developments.

**Appendix A: Parameter values in NUCOM-tundra**

**Acknowledgements**

We acknowledge financial support from The Netherlands Organisation for Scientific Research (NWO-ALW, VIDI grant
864.09.014) and thank P. Wang and L. Belelli Marchesini for providing dwarf shrub and graminoid vegetation biomass data, and latent heat flux (evapotranspiration) and soil temperature profile data for the Chokurdakh Tundra Station in North-eastern Siberia. We thank J. L. van de Poel, N. D. Nobel and E. S. Bargeman for their contributions to the initial development of NUCOM-tundra.

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





**Figure 1.** Schematic presentation of NUCOM-tundra, including the main processes simulated in organic and mineral soil layers and the effects of moisture and available nitrogen on vegetation. Air temperature influences many of these processes, including active layer depth, plant growth, evapotranspiration, snowmelt, decomposition, mineralisation and denitrification.
5   The mineral soil is divided into 10 layers of 10 cm height each.



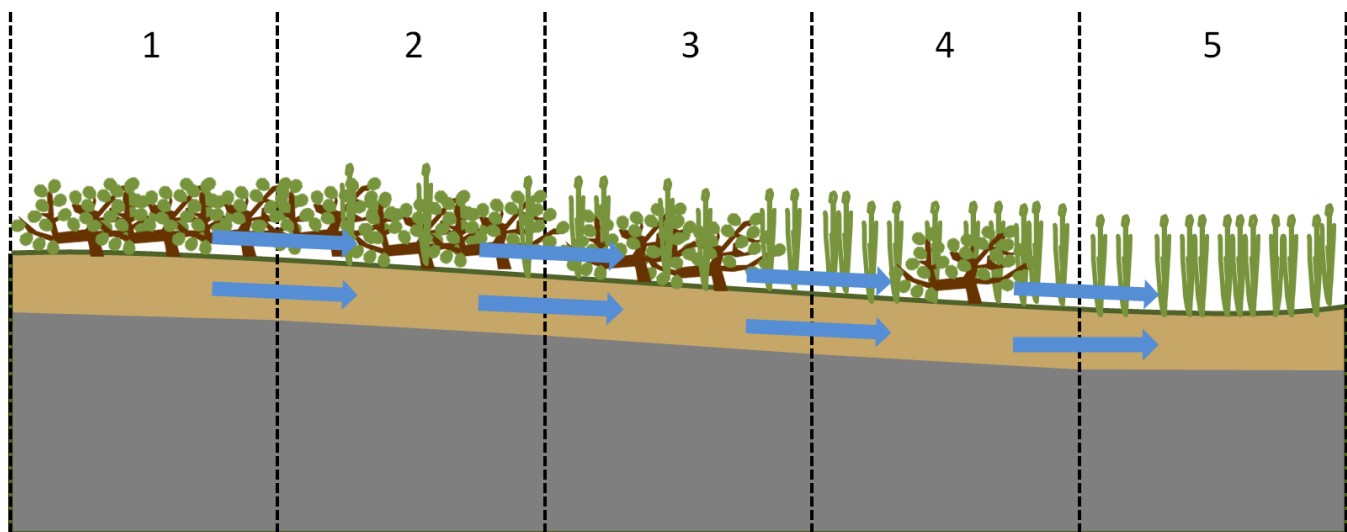

**Figure 2.** Vegetation types used in simulations of NUCOM-tundra. The vegetation types represent landscape positions, ranging from relatively dry shrub dominated to wet graminoid dominated vegetation. Blue arrows indicate water flows.



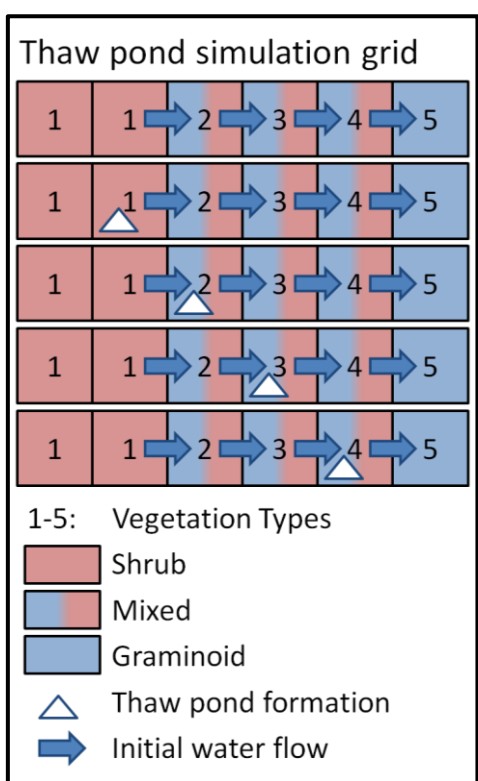

**Figure 3.** Simulation grid used to simulate thaw pond initiation in four different vegetation types. Water flows at the start of the simulation are as indicated in the diagram. After the local permafrost collapse the water flow from the cell with the thaw pond formation to the following downslope cell is stopped (see Figure 6 for the consequences).



**Figure 4.** Simulated summer biomass of moss, graminoids and dwarf shrubs between 1994 and 2074 under different temperature (T, $^0$C) and precipitation (P, mm year$^{-1}$) change scenarios over the 21$^{st}$ century.







**Figure 5.** Simulated summer biomass of moss, graminoids and shrubs averaged over 2065-2074 for sixteen climate scenario simulations in five vegetation types representing a gradient from relatively dry to wet conditions. Temperature and precipitation changes were based on 21$^{st}$ century RCP climate change scenarios. Biomass is total biomass (above and belowground).



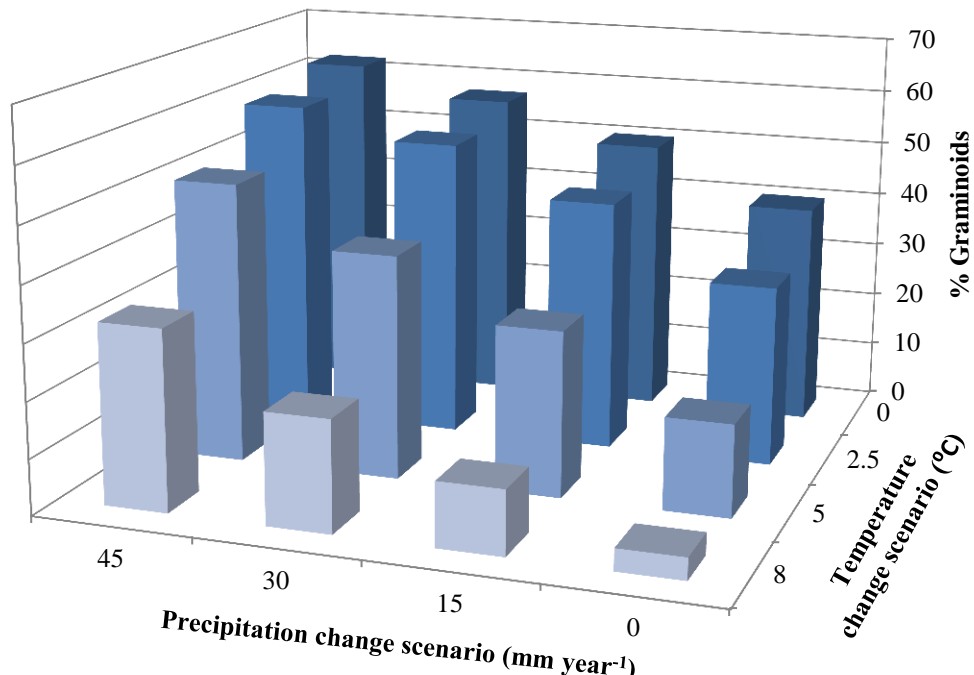

**Figure 6.** Percentage of graminoid biomass in the vascular plant community biomass, averaged over 2064-2074, for vegetation type 3 (initial mixed moist vegetation) for sixteen temperature and precipitation scenarios for the 21st century.




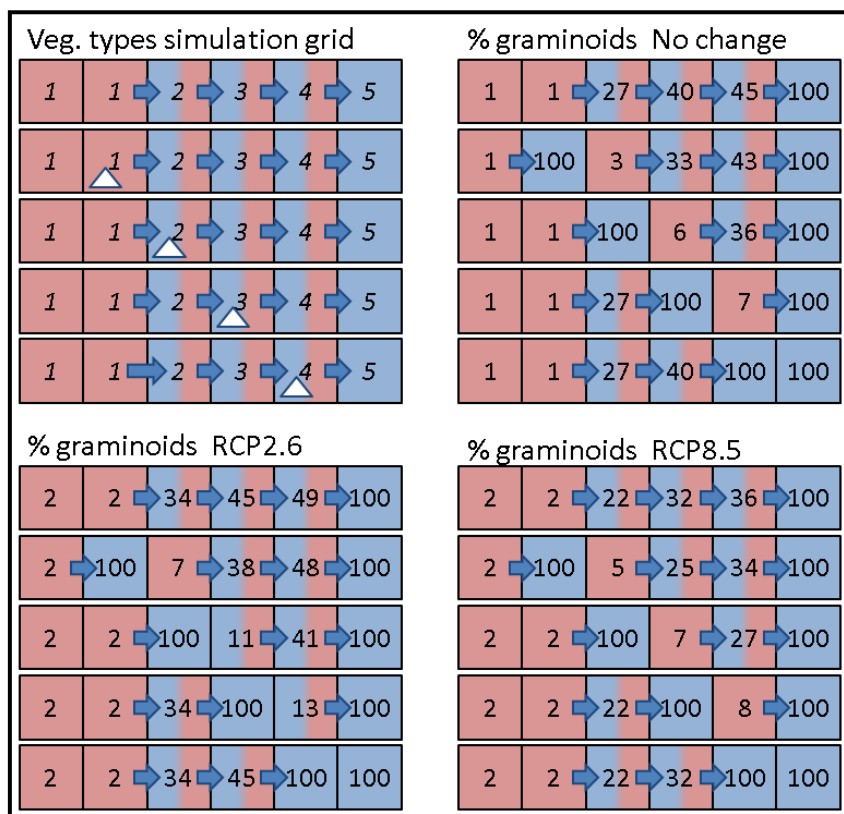

**Figure 7.** Results of thaw pond simulations: simulation settings are in the top left, in which vegetation types (numbers), initial water flows (arrows) and permafrost collapse sites are indicated. The other grids show the percentage of graminoids in total vascular plant biomass during the summers of 2065-2074 and the altered water flows for No change, RCP2.6 and RCP8.5 climate scenarios. Red cells indicate shrub dominated, blue graminoid dominated and both blue and red mixed vegetation.




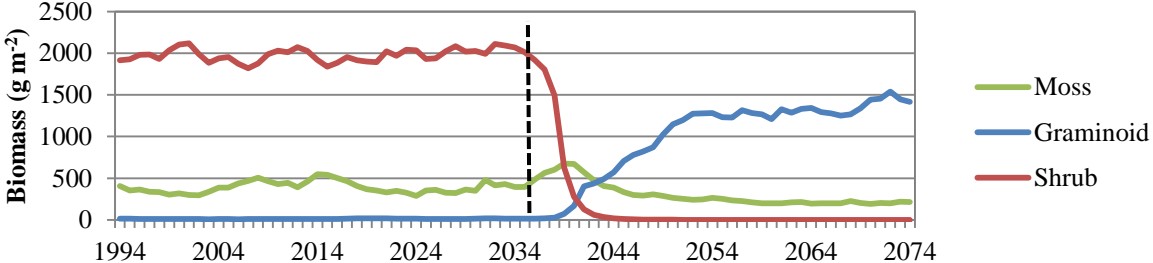

**Figure 8.** Biomass of moss, graminoids and shrubs during a thaw pond formation simulation in an initially dry shrub dominated vegetation type (vegetation type 1) under the No change scenario. The interrupted line indicates the moment of ice wedge collapse.



**Table 1.** Simulated growth-limiting factors for the vascular plant PFTs in vegetation type 3 under different climate change scenarios. Growing season length, soil moisture and light, moisture and nutrient conditions, all during the growing season, were averaged for years 2064 – 1974.

| T change[a] (°C) | P change[a] (mm year⁻¹) | Growing Season Length[b] (days) | Soil Moisture (vol. %) | Graminoids | | | Shrubs | | |
|---|---|---|---|---|---|---|---|---|---|
| | | | | Light[c] (%) | Moisture[d] (%) | Nutrients[e] (%) | Light[c] (%) | Moisture[d] (%) | Nutrients[e] (%) |
| 0 | 0 | 101 | 57.8 | 7.2 | 79.4 | 84.3 | 15.7 | 48.6 | 41.6 |
| 0 | 15 | 101 | 59.2 | 10.3 | 82.6 | 80.9 | 11.2 | 45.5 | 44.9 |
| 0 | 30 | 101 | 60.2 | 12.3 | 84.4 | 79.3 | 8.1 | 41.7 | 46.6 |
| 0 | 45 | 101 | 61.1 | 13.9 | 85.9 | 78.2 | 5.4 | 38.2 | 49.3 |
| 2.5 | 0 | 115 | 52.8 | 6.5 | 68.6 | 83.2 | 28.5 | 61.5 | 33.1 |
| 2.5 | 15 | 115 | 55.9 | 11.7 | 73.7 | 77.2 | 20.6 | 52.2 | 36.6 |
| 2.5 | 30 | 115 | 57.2 | 15.1 | 75.4 | 74.2 | 15.7 | 48.6 | 38.7 |
| 2.5 | 45 | 115 | 58.1 | 17.1 | 77.9 | 72.7 | 13.0 | 46.4 | 40.3 |
| 5 | 0 | 128 | 46.1 | 0.6 | 47.1 | 100 | 47.1 | 69.4 | 33.5 |
| 5 | 15 | 128 | 49.8 | 5.9 | 59.3 | 87.1 | 37.9 | 66.1 | 32.1 |
| 5 | 30 | 128 | 53.4 | 11.1 | 69.4 | 79.7 | 30.7 | 58.5 | 33.4 |
| 5 | 45 | 128 | 55.6 | 16.6 | 72.9 | 74.9 | 23.4 | 51.6 | 35.9 |
| 8 | 0 | 140 | 40.1 | 0.1 | 39.7 | 100 | 61.4 | 70.4 | 36.7 |
| 8 | 15 | 140 | 43.9 | 0.2 | 43.4 | 100 | 58.2 | 64.2 | 36.8 |
| 8 | 30 | 140 | 46.9 | 2.6 | 50.9 | 95.6 | 51.3 | 67.0 | 33.8 |
| 8 | 45 | 140 | 50.6 | 9.3 | 62.2 | 83.9 | 42.4 | 64.9 | 32.9 |

5   a) Temperature and Precipitation change scenario over the 21st century

b) Growing season length is defined as the number of days with mean temperature above 0 °C

c) Percentage of incoming light absorbed during the growing season

d) Percentage of time at which soil moisture conditions were optimal for growth during the growing season (100% = optimal soil moisture conditions during the whole growing season)

10   e) Percentage of realised potential growth (100% = no nutrient limitation)



**Table A1.** Plant Functional Type parameter values in NUCOM-tundra.

| Parameter | Description | Unit | Moss | Gram. | Shrub | Source |
|---|---|---|---|---|---|---|
| $BD_m$ | bulk density moss | $g\ m^{-3}$ | 15660 | | | Heijmans et al., 2004 |
| maxheight | max. height of moss layer | $m$ | 0.045 | | | Blok et al., 2010 |
| minheight | min. height of moss layer | $m$ | 0.005 | | | |
| minleafarea | minimum leaf area | $m^2 m^{-2}$ | | 0.001 | 0.001 | |
| Kext | light extinction coefficient | - | | 0.5 | 0.6 | Heijmans et al., 2008 (Gram.) |
| SLA | Specific Leaf Area | $m^2 g^{-1}$ | | 0.0060 | 0.0139 | Shaver & Chapin, 1991 |
| B | rooting depth coefficient | - | | 0.938 | 0.850 | Murphy et al., 2009 (Gram.); van Wijk, 2007 (Shrub) |
| SRL | specific root length | $m\ g^{-1}$ | | 37.5 | 141.0 | Eissenstat et al., 2000 (Gram.); Pettersson et al., 1993 (Shrub) |
| Gmax | maximum growth | $g\ m^{-2}\ day^{-1}$ | 12.0 | 29.0 | 32.5 | calibrated |
| seedbiom | daily seed biomass input | $g\ m^{-2}\ day^{-1}$ | 0.000033 | 0.000033 | 0.000033 | |
| $w_{min}$ | min. vol. soil water content for growth | % | 20 | 20 | 0 | |
| $w_{low}$ | lowest vol. soil water content for optimal growth | % | 40 | 40 | 10 | |
| $w_{high}$ | highest vol. soil water content for optimal growth | % | 70 | 70 | 65 | |
| $w_{max}$ | max. vol. soil water content for growth | % | 70 | 70 | 70 | |
| $T_{min}$ | minimum air T for growth | $^oC$ | 0 | 1 | 1 | |
| $T_{low}$ | lowest air T for optimal growth | $^oC$ | 3 | 4 | 4 | |
| $N_{min}$ | min. N concentration for growth | $g\ N\ g^{-1}$ | 0.0102 | 0.0122 | 0.0172 | Hobbie, 1996 |
| $N_{max}$ | max. N concentration for N uptake | $g\ N\ g^{-1}$ | 0.0184 | 0.0352 | 0.0278 | Hobbie, 1996 |
| Kallor | Allocation of growth to fine roots | - | | 0.30 | 0.33 | calibrated (Gram.); Shaver and Chapin, 1991 (Shrub) |
| Kallos | allocation growth towards stems | - | | 0.24 | 0.14 | calibrated (Gram.); Shaver and Chapin, 1991 (Shrub) |
| Kallol | allocation growth towards leaves | - | | 0.46 | 0.53 | calibrated (Gram.); Shaver and Chapin, 1991 (Shrub) |
| Krear | reallocation of N in roots to storage | - | | 0.30 | 0.10 | Heijmans et al., 2008 |
| Kreas | reallocation of N stems to storage | - | | 0 | 0 | Heijmans et al., 2008 |
| Kreal | reallocation of N leaf to storage | - | | 0.34 | 0.25 | Chapin et al., 1975 (Graminoid) |
| $mort_{moss}$ | mortality moss | $day^{-1}$ | 0.000914 | | | Chapin et al., 1996 |
| $mort_r$ | mortality root | $day^{-1}$ | | 0.000825 | 0.001015 | calibrated |
| $mort_s$ | mortality stems | $day^{-1}$ | | 0.000232 | 0.000067 | calibrated |
| $mort_l$ | mortality leaf | $day^{-1}$ | | 0.007315 | 0.010790 | calibrated |
| $maxmort_s$ | maximum mortality stems | $day^{-1}$ | | 0.002500 | 0.002500 | |
| kdec_m | decomposition moss litter | $day^{-1}$ | 0.001560 | | | Lang et al., 2009 |
| kdec_r | decomposition root litter | $day^{-1}$ | | 0.001691 | 0.000946 | Heal and French, 1974 |
| kdec_s | decomposition stems litter | $day^{-1}$ | | 0.000758 | 0.001256 | Heal and French, 1974 |
| kdec_l | decomposition leaf litter | $day^{-1}$ | | 0.001564 | 0.002167 | Hobbie and Gough, 2004 |




**Table A2.** Parameter values related to the soil profile, microbial processes and hydrology in NUCOM-tundra.

| Parameter | Description | Unit | Value | Source |
|---|---|---|---|---|
| maxlayer | number of mineral layers | - | 10 | |
| layerdepth | thickness of mineral layer | $m$ | 0.10 | |
| $BD_{org}$ | organic matter density | $g\ m^{-3}$ | 150000 | Marion and Miller, 1982 |
| Nfixation_m | nitrogen fixation rate (by moss) | $g\ N\ day^{-1}$ | 0.00025 | |
| Ndeposition | nitrogen deposition rate | $g\ N\ day^{-1}$ | 0.00027 | |
| NCcrit | critical N:C ratio for mineralization | $g\ N\ g\ C^{-1}$ | 0.008 | |
| $e^{asseff}$ | microbial assimilation efficiency | - | 0.2 | Heijmans et al., 2008 |
| $\alpha$ | optimal denitrification rate | $day^{-1}$ | 0.1 | Heinen, 2006 |
| $S_{tres}$ | minimum fraction of water filled pores for denitrification | | 0.9 | |
| $S_{max}$ | fraction of water filled pores for max. denitrification | | 1 | |
| $Q_{10}$ | denitrification increase factor with 10°C increase | | 2 | Heinen, 2006 |
| $T_{ref}$ | reference temperature (for denitrification) | $^oC$ | 20 | Heinen, 2006 |
| DDF | Degree Day Factor for snowmelt | $mm\ ^oC^{-1}\ day^{-1}$ | 5.3 | Lundberg & Beringer, 2005 |
| $FieldCap_{org}$ | vol. water content at field capacity in organic layer | $mm\ mm^{-1}$ | 0.36 | Zotarelli et al., 2010 |
| $MaxCap_{org}$ | max. vol. water content in organic layer (saturation) | $mm\ mm^{-1}$ | 0.70 | |
| $FieldCap_{min}$ | vol. water content at field capacity in mineral layer | $mm\ mm^{-1}$ | 0.42 | Saxton and Rawls, 2006 |
| $MaxCap_{min}$ | max. vol. water content in mineral layer (saturation) | $mm\ mm^{-1}$ | 0.50 | Saxton and Rawls, 2006 |
| runoffr | surface water runoff | $day^{-1}$ | 0.10 | |
| interflowr | Lateral drainage of water through organic layer | $day^{-1}$ | 0.01 | |
| evaporation | fraction evaporation of total evapotranspiration | - | 0.5 | |
| evapodepth | Soil depth over which evaporation occurs | $cm$ | 10 | |
| wheight | height of the upper soil layer for moisture calculation | $cm$ | 10 | |
| leachr | fixed nitrogen leach rate to deeper soil layer | $day^{-1}$ | 0.00163 | |
| $leachr_{max}$ | maximum nitrogen leach rate | $day^{-1}$ | 0.00747 | |