# Peer review of "Simulating the effects of temperature and precipitation change on vegetation composition in Arctic tundra ecosystems"

_Biogeosciences, 2016_

## Referee Comment (RC1) · Anonymous Referee #1 · 1 Jun 2016

This paper is well written and logically structured. It presents a study with many potentially interesting insights on the High Arctic permafrost ecosystem dynamics (vegetation competition and succession) in response to future climate change, permafrost thawing, and lateral interaction in hydrology and thermokarst development. However, some major mechanisms behind the processes associated with the interaction between biotic and abiotic factors haven't been clearly demonstrated.

I suggest the issues the paper should address in the following phase. (1) Nutrient availability and mobility. The N availability is determined by the rate of minimization and fixation of N in response to the extent of climate changes. Their net effects determine the nutrient constraint for different vegetation species. In addition, snow is another

important aspect to influence the subsurface temperature and then the N cycling. The N mobility can be reflected by how dry ecosystems interacts wet ecosystems through water movement. These two issues have not been well investigated in the current modelling work, but they are fundamental in understanding how growth of plant function types are influenced by environmental changes.

(2) The model needs a thorough evaluation in the performance of simulating soil water, evapotranspiration and soil temperature, active layer depth, water table depth for the period the observations are available. This is the basis to convince the readers to believe the efficiency of the model. Particularly, the simulated soil temperature doesn't look correct in the 40, 80 cm.

(3) I suggest use the percentage of increase to indicate the change of precipitation. For this study site, 45 mm/year, (i.e. 20% increase of annual precipitation) seems much lower than the IPCC CMIP5 prediction for the RCP8.5 scenario. For instance, http://www.nature.com/nature/journal/v509/n7501/pdf/nature13259.pdf

Other minor issues:

The rate of biomass increase is suggested to use the unit "g m-2 yr-1".

How the development stages of thawing pond are evolved in different climate scenarios is suggested to demonstrate. For instance, the time series of water table depth in climate scenario runs.

The title should be catchier. The current one seems quite broad.

---

## Referee Comment (RC2) · Anonymous Referee #2 · 24 Jul 2016

This is an interesting and relevant studied, which in my opinion deserves to be published in Biogeosciences. The paper presents in a clear and interesting way potential changes of Arctic tundra under warming/precipitation change/permafrost thaw. Especially, addressing 3 factors in combination, i.e. warming, precipitation and permafrost thaw is a relevant contribution to our understanding of tundra change.

The paper is well written and the results are clearly presented. As this study represents a modelling approach, I would find it helpful if some modelling related issues could be clarified. In particular, many parameters in the NUCOM-tundra model were defined based on e.g. vegetation composition found in the field, so I was sometimes uncertain what I learned in the paper about mechanisms responsible for changes in the tundra.

[Figure]

Also related to this issue: of course simplifications/assumptions need to be made for a model, especially if access to measured data is limited. However, I asked myself a few times if the simplification were justified.

A few examples. Abstract. L.24. The simulations suggest that shrubs are better light competitors... etc. If I understand the model right, shrubs are good competitors because they were defined as good competitors in the first place. Not that this would be incorrect. But several times I get the impression that findings are not necessarily a result of the model but a result of how the model was set up, which assumptions were made and which data were used to feed the model. Again, this is certainly an issue that can be said for all models. But I think the text needs some rephrasing to be clear about what is indeed a model outcome (e.g. increase of graminoids under wetter conditions) and what is not. To me the text seems to go too far, which mechanisms can actually explained by this model and which cannot. See related comments below.

Questionable assumption? p4 l21. Graminoids and dwarf shrubs are assumed to be equally tall. The authors may have their reasons to do so, but this is not entirely clear to me. Betula nana can grow easily 2.5 m tall (e.g. in parts of Alaska) and arctic graminoids don't. An incorrect assumption here could have a large influence on the results.

Explaining mechanisms? P. 9 l16ff. The authors state that the NUCOM model was developed to assess which mechanisms are responsible for tundra change. I found this statement somewhat questionable because many very important mechanisms remain unknown when assumptions are made for models. The biomass example above is one such example. The issue that rooting depth in a warmer climate is not known is another example, but discussed later in the discussion. It might be helpful if the authors adapt their wording a bit. E.g. that they refer mechanisms to effects of warming vs. precipitation, which is the novel contribution of this paper.

So I suggest that the authors go through the entire manuscript another time and rethink carefully how to not over-sell their results. Apart from that, to state this again, I like the paper and find it helpful and novel.

---

## Author Comment (AC1) · 18 Aug 2016

**Author comment on "Simulating the effects of temperature and precipitation change on vegetation composition in Arctic tundra ecosystems" by H. van der Kolk et al.**

Reply to comments by anonymous Referee #1

*This paper is well written and logically structured. It presents a study with many potentially interesting insights on the High Arctic permafrost ecosystem dynamics (vegetation competition and succession) in response to future climate change, permafrost thawing, and lateral interaction in hydrology and thermokarst development. However, some major mechanisms behind the processes associated with the interaction between biotic and abiotic factors haven't been clearly demonstrated.*

REPLY: We thank the referee for his/her constructive comments. We appreciate that the referee recognizes that the model has the potential to provide interesting insights in the response of tundra vegetation composition to climate change, gradual permafrost thawing and abrupt permafrost thaw (thermokarst). In our study the focus is on the vegetation modelling and we acknowledge that for the abiotic factors there is room for improvement in the model. In the next phase of model development we plan to couple the vegetation model to a methane emission/soil physics model (e.g. Mi et al. 2014).

*I suggest the issues the paper should address in the following phase.*

*(1) Nutrient availability and mobility. The N availability is determined by the rate of minimization and fixation of N in response to the extent of climate changes. Their net effects determine the nutrient constraint for different vegetation species. In addition, snow is another important aspect to influence the subsurface temperature and then the N cycling. The N mobility can be reflected by how dry ecosystems interacts wet ecosystems through water movement. These two issues have not been well investigated in the current modelling work, but they are fundamental in understanding how growth of plant function types are influenced by environmental changes.*

REPLY: In general, growth of tundra plants is assumed to be limited by nitrogen availability and therefore simulation of nitrogen dynamics is essential for a dynamic tundra vegetation model. In our model, mineralisation of soil organic nitrogen is temperature-dependent, so will respond to climatic changes. In addition, atmospheric nitrogen deposition, nitrogen fixation, nitrogen leaching to deeper soil layers and denitrification of nitrogen are included in the model. The available nitrogen in the different soil layers influences competition between the deeper-rooted graminoids and shallower-rooted dwarf shrubs. Indeed, snow influences the winter soil temperatures, which is not explicitly included in our regression-based soil temperature simulation, see further reply to comment 2. However, at our study site with an extremely harsh winter climate nitrogen mineralisation is always extremely low in winter. In summer, water movement can indeed transport nitrogen from the elevated shrub patch to the sedge-dominated depression. This was not included in the current version of the model, but will be implemented in the model for the revised manuscript.

*(2) The model needs a thorough evaluation in the performance of simulating soil water, evapotranspiration and soil temperature, active layer depth, water table depth for the period the observations are available. This is the basis to convince the readers to believe the efficiency of the model. Particularly, the simulated soil temperature doesn't look correct in the 40, 80 cm.*

REPLY: Unfortunately we have no time series of observations for soil moisture, active layer depth and water table depth for a thorough comparison with model outcome. Indeed, the simulated soil temperature at 40 and 80 cm depth does not look good in winter. It has been acknowledged in the

text that the soil temperature model is based on temperatures in the growing season only, as the below-zero temperature in winter hardly affects the model output (Supplement S1.3.4). We used a regression approach which is based on air temperature and does not take snow characteristics and phase changes into account. Still, realistic temperatures for the upper (most important) soil layers in summer have been simulated, but there are deviations for winter temperatures (Supplement Figure S1.3). In the next phase of model development, we plan to couple the vegetation model to a more advanced soil physical model.

Soil moisture and water table depth are extremely variable even at a small spatial scale in the tundra landscape. The wet depressions (resembling the wettest vegetation type in the model) are much wetter compared to the slightly elevated shrub patches. Therefore, we choose to simulate different vegetation types along a water movement gradient from relatively well-drained elevated shrub patches to downstream sedge-dominated depressions receiving water from surrounding. As described in the manuscript, the model simulates a build-up of the snow layer in winter, followed by a wet period in spring following snowmelt. In summer, the soil gradually dries as evapotranspiration (derived from measured latent heat flux) generally exceeds precipitation, the extent of drying depending on the amount of precipitation and temperature. This sequence and dependence on precipitation matches with the observations at our study site in north-eastern Siberian tundra.

Unfortunately, only limited soil moisture observations are available for the Kytalyk field research site. Soil moisture measurements in graminoid, mixed and shrub dominated vegetation types indicate clear trends that moisture during the growing season is high in graminoid but low in shrub dominated sites, similar as is simulated in the model. Soil temperature in the upper soil layers is in the range of the simulated values. For deeper layers, the simulated soil temperatures are higher than observed, resulting in an active layer that is thicker than observed. We will add a more thorough evaluation of measured and simulated soil moisture, soil temperature and active layer thickness values in the revised manuscript.

*(3) I suggest use the percentage of increase to indicate the change of precipitation. For this study site, 45 mm/year, (i.e. 20% increase of annual precipitation) seems much lower than the IPCC CMIP5 prediction for the RCP8.5 scenario. For instance, http://www.nature.com/nature/journal/v509/n7501/pdf/nature13259.pdf*

REPLY: We accept this suggestion and will use the percentage instead of the actual precipitation increase. Indeed, the precipitation increases used for the current simulations are too low. These values will be corrected in the model simulations and the revised manuscript.

*Other minor issues:*

*The rate of biomass increase is suggested to use the unit "g m-2 yr-1".*

REPLY: Suggestion accepted

*How the development stages of thawing pond are evolved in different climate scenarios is suggested to demonstrate. For instance, the time series of water table depth in climate scenario runs.*

REPLY: Suggestion accepted, we will add water table depth to the thaw pond scenario graphs in the revised manuscript.

*The title should be catchier. The current one seems quite broad.*

REPLY: We agree that the title "Simulating the effects of temperature and precipitation change on vegetation composition in Arctic tundra ecosystems" could be catchier. The new title will be

"Potential Arctic tundra vegetation shifts in response to changing temperature, precipitation and permafrost thaw"

Or "A new Arctic tundra vegetation model for simulating vegetation shifts due to climate change and permafrost thaw"

---

## Author Response (AR1)

**Final response to comments on "Simulating the effects of temperature and precipitation change on vegetation composition in Arctic tundra ecosystems" by H. van der Kolk et al.**

Reply to comments by anonymous Referee #1

*This paper is well written and logically structured. It presents a study with many potentially interesting insights on the High Arctic permafrost ecosystem dynamics (vegetation competition and succession) in response to future climate change, permafrost thawing, and lateral interaction in hydrology and thermokarst development. However, some major mechanisms behind the processes associated with the interaction between biotic and abiotic factors haven't been clearly demonstrated.*

REPLY: We thank the referee for his/her constructive comments. We appreciate that the referee recognizes that the model has the potential to provide interesting insights in the response of tundra vegetation composition to climate change, gradual permafrost thawing and abrupt permafrost thaw (thermokarst). In our study the focus is on the vegetation modelling and we acknowledge that particularly for the abiotic factors there is room for improvement in the model. In the next phase of model development we plan to couple the vegetation model to a soil physics/methane emission model (e.g. Mi et al. 2014).

*I suggest the issues the paper should address in the following phase.*

*(1) Nutrient availability and mobility. The N availability is determined by the rate of minimization and fixation of N in response to the extent of climate changes. Their net effects determine the nutrient constraint for different vegetation species. In addition, snow is another important aspect to influence the subsurface temperature and then the N cycling. The N mobility can be reflected by how dry ecosystems interacts wet ecosystems through water movement. These two issues have not been well investigated in the current modelling work, but they are fundamental in understanding how growth of plant function types are influenced by environmental changes.*

REPLY: In general, growth of tundra plants is assumed to be limited by nitrogen availability and therefore simulation of nitrogen dynamics is essential for a dynamic tundra vegetation model. In our model, mineralisation of soil organic nitrogen is temperature-dependent, so will respond to climatic changes. The soil temperature dependence of nitrogen mineralisation is now explicitly mentioned on page 5, lines 7-8. In addition, atmospheric nitrogen deposition, nitrogen fixation, nitrogen leaching to deeper soil layers and denitrification of nitrogen are included in the model. The available nitrogen in the different soil layers influences competition between the deeper-rooted graminoids and shallower-rooted dwarf shrubs.

Indeed, snow influences the winter soil temperatures, which is not explicitly included in our regression-based soil temperature simulation, see further reply to comment 2. However, at our study site with an extremely harsh winter climate nitrogen mineralisation is always extremely low in winter.

In summer, water movement can indeed transport nitrogen from the elevated shrub patch to the sedge-dominated depression. This has been included in the model for the revised manuscript, and is mentioned on page 5, line 13 and described in S1.3.2. This nitrogen transport with water movement and the larger precipitation increase in the climate scenarios (comment 3) changed the results of the simulations, see further reply to comment 3.

*(2) The model needs a thorough evaluation in the performance of simulating soil water, evapotranspiration and soil temperature, active layer depth, water table depth for the period the observations are available. This is the basis to convince the readers to believe the efficiency of the model. Particularly, the simulated soil temperature doesn't look correct in the 40, 80 cm.*

REPLY: Indeed, the simulated soil temperature at 40 and 80 cm depth does not look good in winter. It has been acknowledged in the text that the soil temperature model is based on temperatures in the growing season only, as the below-zero temperature in winter hardly affects the model output (Supplement S1.3.4). We used a regression approach which is based on air temperature and does not take snow characteristics and phase changes into account. Still, realistic temperatures for the upper (most important for N mineralisation) soil layers in summer have been simulated, but there are (large) deviations for winter temperatures (Supplement Figure S1.3). These deviations did not affect the simulated periods of frozen and thawed soil, which were in agreement with the observations (Supplement page 29, lines 6-8). For broader applications of the model it will be necessary to couple the vegetation model to a more advanced soil physical model (page 11, lines 29-30). We added a new supplement section (S4.2) on the evaluation of soil temperature and active layer thickness to the revised manuscript. In the main text, the comparison with the soil temperature time series is mentioned on page 6, lines 14-15. The results of this evaluation have been summarized on page 8, lines 6-13.

New supplement section S4.2 also evaluates the simulated active layer thickness (ALT). Based on measured thawing depths on 20 dates, it appeared that the simulated and measured ALT were strongly correlated (new figure S4.2), but simulated ALT proved to be slightly too deep: average difference between simulated and measured ALT was 14 cm. Next, we assessed the consequences of too deep ALT for the simulated vegetation composition in a simulation where deep soil temperatures (and thus active layer thickness) were manually set back. The result is that too deep simulated ALT is to the advantage of graminoids (new figure S4.3). This can be explained by the fact that our vegetation model includes differential rooting depths for graminoids and shrubs. Graminoids are more capable of acquiring available nutrients in deeper soil layers, as they are deeper-rooted than the dwarf shrubs. The result implies that simulated graminoid biomass may be overestimated as a consequence of overestimated ALT. This is discussed in a new paragraph on page 11, lines 19-30. Although there is uncertainty regarding the simulated ALT and the amount of nutrients becoming available, the pattern of vegetation shifts in response to temperature and precipitation changes remained the same.

Unfortunately we have no time series of observations for soil moisture and water table depth for a thorough comparison with model outcome. The meteo tower at the Kytalyk field station does not measure soil moisture. Soil moisture and water table depth are extremely variable even at a small spatial scale in the tundra landscape. The wet depressions (resembling the wettest vegetation type in the model) are much wetter compared to the slightly elevated shrub patches. Therefore, we choose to simulate different vegetation types along a water movement gradient from relatively well-drained elevated shrub patches to downstream sedge-dominated depressions receiving water from their surroundings. Limited soil moisture measurements in graminoid, mixed and shrub dominated vegetation types (Wang et al. 2016) indicate clear trends that moisture during the growing season is high in graminoid but low in shrub dominated sites, similar as is simulated in the model. As described in the manuscript, the model simulates a build-up of the snow layer in winter, followed by a wet period in spring following snowmelt. In summer, the soil gradually dries as evapotranspiration (derived from measured latent heat flux) generally exceeds precipitation, the extent of drying depending on the amount of precipitation and temperature. This sequence and dependence on

precipitation matches with our personal observations at the study site in north-eastern Siberian tundra. We do have several point measurements of soil moisture for several years. However, these measurements are difficult to compare with the simulated soil moisture values as they represent different soil layers. Measurements using a Thetaprobe were taken from the top soil layer, i.e. the top 10cm including moss which easily dries out. Simulated soil moisture values represent the whole organic soil layer (which is 15-20cm deep). Therefore, the measured values show much more variation than the simulated values. Consequently, we could not reliably compare measured and simulated soil moisture data.

*(3) I suggest use the percentage of increase to indicate the change of precipitation. For this study site, 45 mm/year, (i.e. 20% increase of annual precipitation) seems much lower than the IPCC CMIP5 prediction for the RCP8.5 scenario. For instance, http://www.nature.com/nature/journal/v509/n7501/pdf/nature13259.pdf*

REPLY: We accepted this suggestion and used the percentage instead of the actual precipitation increase, which resulted in larger increases of precipitation in the climate scenarios and changed the outcome of the simulations. For the main climate change scenarios (RCP2.6, Intermediate, RCP8.5) graminoids performed relatively well in comparison to previous simulation. This improvement of graminoid performance can be attributed to increased precipitation in climate change scenarios (increased soil moisture content during growing season). As a consequence of these stronger precipitation effects, we highlighted the role of precipitation earlier in the discussion (page 11, lines 6-18) and had to adapt some text in the results (page 8, lines 31-32; page 9, lines 1-3).

*Other minor issues:*

*The rate of biomass increase is suggested to use the unit "g m-2 yr-1".*

REPLY: Sorry, it is not clear to us to which figure or part of the text the referee is referring to. Figures show biomass expressed in g m$^{-2}$.

*How the development stages of thawing pond are evolved in different climate scenarios is suggested to demonstrate. For instance, the time series of water table depth in climate scenario runs.*

REPLY: Suggestion accepted, we added the soil moisture content over time as an extra line to the thaw pond scenario graphs in the revised manuscript (Fig. ...).

*The title should be catchier. The current one seems quite broad.*

REPLY: We agree that the title "Simulating the effects of temperature and precipitation change on vegetation composition in Arctic tundra ecosystems" could be catchier. We changed the title to:

"Potential Arctic tundra vegetation shifts in response to changing temperature, precipitation and permafrost thaw"

An alternative could be: "A new Arctic tundra vegetation model for simulating vegetation shifts due to climate change and permafrost thaw"

Reply to comments by anonymous Referee #2

*This is an interesting and relevant studied, which in my opinion deserves to be published in Biogeosciences. The paper presents in a clear and interesting way potential changes of Arctic tundra under warming/precipitation change/permafrost thaw. Especially, addressing 3 factors in combination, i.e. warming, precipitation and permafrost thaw is a relevant contribution to our understanding of tundra change.*

REPLY: We thank the referee for his/her constructive comments. We appreciate that the referee recognizes that our study has the potential to provide interesting insights in the response of tundra vegetation composition to climate change, gradual permafrost thawing and abrupt permafrost thaw.

*The paper is well written and the results are clearly presented. As this study represents a modelling approach, I would find it helpful if some modelling related issues could be clarified. In particular, many parameters in the NUCOM-tundra model were defined based on e.g. vegetation composition found in the field, so I was sometimes uncertain what I learned in the paper about mechanisms responsible for changes in the tundra.*

REPLY: It is correct that some of the parameter values for the model have been derived from field measurements at our study site in north-eastern Siberian tundra, see further reply to comments below. From our field studies we have gained insights in the functioning of (dwarf) shrubs and graminoids which have been used for the model. For example, the graminoids are deeper-rooted than the shrubs which implies that the graminoids are better competitors for nutrients in the deeper soil layers. The model synthesizes own findings from field studies with knowledge on Arctic tundra ecosystem functioning from literature.

*Also related to this issue: of course simplifications/assumptions need to be made for a model, especially if access to measured data is limited. However, I asked myself a few times if the simplification were justified.*

*A few examples. Abstract. L.24. The simulations suggest that shrubs are better light competitors... etc. If I understand the model right, shrubs are good competitors because they were defined as good competitors in the first place. Not that this would be incorrect. But several times I get the impression that findings are not necessarily a result of the model but a result of how the model was set up, which assumptions were made and which data were used to feed the model. Again, this is certainly an issue that can be said for all models. But I think the text needs some rephrasing to be clear about what is indeed a model outcome (e.g. increase of graminoids under wetter conditions) and what is not. To me the text seems to go too far, which mechanisms can actually explained by this model and which cannot. See related comments below.*

REPLY: Of course a result of a model depends on how the model was set up, etc. This has the advantage that it is possible to trace back how a certain model outcome has been generated and provides a potential mechanism for the simulated vegetation change. It is correct that the mechanism is not independent of how the model was build and we agree that this should not be overstated. Still, we think that models are a helpful tool for exploring mechanisms of vegetation change.

We disagree that shrubs were defined as good competitors in the first place. Several plant traits (parameters values) are related to competition for light or nutrients. For light competition, leaf area and light extinction are important, which are influenced by parameters as specific leaf area, biomass allocation to leaves, leaf mortality and light extinction coefficient. Similarly, nutrient competition is influenced by parameters as specific root length, root distribution and nutrient requirements. This explanation is added to page 12, lines 17-23. Values for the parameters have been assigned independently, partly based on literature values and partly derived from own measurements. It is the combination of multiple plant traits and environmental conditions that determines which plant type is a good competitor. Based on our field observations, we assumed the (dwarf) shrubs and graminoids to be equally tall in the model. Therefore, the shrubs were not a-priori the better light competitor. We removed the sentence about shrubs being better light competitors (page 1, line 27), as it is not that clear from our simulations anyway.

*Questionable assumption? p4 l21. Graminoids and dwarf shrubs are assumed to be equally tall. The authors may have their reasons to do so, but this is not entirely clear to me. Betula nana can grow easily 2.5 m tall (e.g. in parts of Alaska) and arctic graminoids don't. An incorrect assumption here could have a large influence on the results.*

REPLY: Based on observations at our study site in north-eastern Siberian tundra we assumed the graminoids (*Eriophorum* spp.) and dwarf shrubs (*Betula nana*) to be equally tall, which is a simplification, but not unrealistic. Also in Alaska, in moist tussock tundra as in Toolik Lake, dwarf shrubs and graminoids are equally tall (Heijmans, personal observations). We acknowledge that in some other tundra areas the shrubs, including *B. nana*, can grow taller than the graminoids, although this may be limited to places with some additional nutrient supply. For a wider application of the vegetation model, i.e. to include transitions to tall shrub vegetation, a variable plant height up to a maximum would be required. We added this discussion to page 12, lines 27-31

*Explaining mechanisms? P. 9 l16ff. The authors state that the NUCOM model was developed to assess which mechanisms are responsible for tundra change. I found this statement somewhat questionable because many very important mechanisms remain unknown when assumptions are made for models. The biomass example above is one such example. The issue that rooting depth in a warmer climate is not known is another example, but discussed later in the discussion. It might be helpful if the authors adapt their wording a bit. E.g. that they refer mechanisms to effects of warming vs. precipitation, which is the novel contribution of this paper.*

REPLY: The focus of this modelling study was on exploring potential vegetation shifts in response to changes in temperature, precipitation and permafrost thaw, as is evident from the abstract. We agree that in parts of the main text the wording is a bit misleading with too much emphasis on mechanisms. We went through the whole text and adapted the wording in several places (e.g. page 1, lines 15, 18; page 4, line 1; page 10, lines 8-13).

*So I suggest that the authors go through the entire manuscript another time and re-think carefully how to not over-sell their results. Apart from that, to state this again, I like the paper and find it helpful and novel.*

**Potential Arctic tundra vegetation shifts in response to changing temperature, precipitation and permafrost thaw**

[revised manuscript text omitted]

5 ## 4. Discussion

The effects of climate change on tundra vegetation are complex since temperature and precipitation drive changes in hydrology, active layer depths, nutrient availability and growing season length, which interact with each other (Serreze et al., 2000; Hinzman et al., 2005).
10  NUCOM-tundra has been developed to explore future Arctic tundra vegetation composition changes in response to climate scenarios, thereby taking changes in soil moisture, thawing depth and nutrient availability into account.

**4.1 Effects of increases in temperature and precipitation**

15 Our climate scenario simulations suggest a significant increase in biomass with continuing climate change, and  especially increased graminoid abundance under scenarios with different magnitudes of climate change. For modest and strong emission scenarios (% precipitation Intermediate and RCP8.5), both shrub and graminoid biomass increased. These simulations are in line with biomass increases in both shrubs and graminoids that have been observed during the past
20 decades in Arctic tundra landscapes as a response to temperature increase (Dormann and Woodin, 2002; Walker et al., 2006; Hudson and Henry, 2009). Shrub expansion  has been observed in many places in the Arctic (Sturm et al., 2001; Tape et al., 2006; Myers-Smith et al., 2011), but is not explicitly simulated by NUCOM-tundra for the RCP-based climate change scenarios which combine temperature and precipitation increases.

25 As nutrient availability limits plant growth in tundra landscapes (Shaver et al., 2001), a positive effect of warming on nutrient availability is a likely explanation for biomass increase observed in tundra vegetation (Hudson and Henry, 2009). Climate warming might influence nutrient availability positively by lengthening of the growing season, active layer deepening and increased microbial activity. In our simulations, nutrients were especially limiting for the shrubs, likely due to their shallow rooting systems. Compared to shrubs, graminoids root relatively deep (Wang et al. 2016). As a consequence,
30 active layer deepening is expected to favour especially graminoids. It is, however, unclear how plant root morphology

responds to climate warming. An experimental warming study in dry tundra demonstrated that plants do not necessarily root deeper in response to warmer temperatures, but instead may concentrate their main root biomass in the organic layer where most nutrient mineralization takes place (Björk et al., 2007). Nevertheless, it is likely that growing season lengthening and increased microbial mineralization of soil organic matter improve growing conditions for both shrubs and graminoids, as also graminoids have been shown to respond strongly upon fertilization (e.g. Jonasson, 1992).

Strikingly, the climate simulations in this study show that shifts in vegetation composition are not only dependent on temperature change, but are strongly affected by precipitation changes as well. Simulated soil moisture contents decreased with higher temperature and lower precipitation scenarios. Evapotranspiration is an important hydrological process determining soil moisture during the growing season. Throughout the growing season, the top soil layer dries out as evapotranspiration exceeds precipitation during this period, which is in agreement with observations at our study site (Supplement Section S2). Higher summer temperatures increase potential evapotranspiration, and thus lead to drier soils, if precipitation remains unchanged. Consequently, the area of relatively dry sites characterized by dense dwarf shrub vegetation might increase. A similar misbalance of temperature and precipitation change between 1950 and 2002 has been proposed by Riordan et al. (2006) as one of the possible causes for drying of thermokarst ponds in Alaska. A second mechanism for tundra drying with higher temperatures is increased water drainage enabled by gradual deepening of the active layer or permafrost degradation. The latter mechanism is especially important in the discontinuous permafrost zone, where climate change may cause loss of permafrost at thermokarst sites and subsequently lead to increased drainage to adjacent areas (Yoshikawa and Hinzman, 2003).

Whereas NUCOM-tundra predicts that especially graminoids benefit under the main climate change scenarios, other Arctic vegetation models mainly simulate shrub expansion (e.g. Epstein et al., 2000; Euskirchen et al., 2009; Yu et al., 2011). As moisture conditions did not differ much between the combined climate change scenarios (Table 1), the relatively large increase of graminoids in those scenarios is likely enabled by active layer deepening. Nutrients that become available in deeper soil layers are easily accessible by graminoids which are able to root throughout the whole active layer, whereas shrubs rely on the nutrient availability in the top soil layers as their root system is confined to that region. It is important to note that NUCOM-tundra currently uses a simple soil temperature soil module, which calculates soil temperature based on air temperature (Supplement S1.3.4). Especially for deeper soil layers, this module lacks accuracy to precisely predict changes in active layer thickness for different temperature change scenarios, which results in uncertainty in both the depth of the active layer and the amounts of nutrients becoming available. These processes influence graminoid growth more than shrub growth. Coupling NUCOM-tundra to an advanced soil temperature model is needed to simulate the physical soil processes more accurately. The climate simulations in this study show that shifts in vegetation composition are not only dependent on temperature change, but are strongly affected by precipitation changes as well. Simulated soil moisture contents decreased with higher temperature and lower precipitation scenarios. Evapotranspiration is an important

~~hydrological process determining soil moisture during the growing season. Throughout the growing season, the top soil layer dries out as evapotranspiration exceeds precipitation during this period, which is in agreement with observations at our study site (Supplement Section S2). Higher summer temperatures increase potential evapotranspiration, and thus lead to drier soils, if precipitation remains unchanged. Consequently, the area of dry vegetation sites characterized by their dense dwarf shrub coverage might increase. A similar misbalance of temperature and precipitation change between 1950 and 2002 has been proposed by Riordan et al. (2006) as one of the possible causes for drying of thermokarst ponds in Alaska. A second mechanism for tundra drying with higher temperatures is increased water drainage enabled by gradual deepening of the active layer or permafrost degradation. The latter mechanism is especially important in the discontinuous permafrost zone, where climate change may cause loss of permafrost at thermokarst sites and subsequently lead to increased drainage to adjacent areas (Yoshikawa and Hinzman, 2003).~~

Previously, the effectiveness of shrubs to deal with increased nutrient levels has been proposed as an explanation for observed shrub expansion in the Arctic (Shaver et al., 2001; Tape et al., 2006). *Betula nana* responds to higher nutrient availability by increasing its biomass, which is mainly due to increased secondary stem growth (Shaver et al., 2001). However, *Betula nana* is also known to respond to increased temperatures and fertilization by growing taller and by producing more shoots and tillers, thereby increasing its ability to compete for light with other species (Chapin and Shaver, 1996; Hobbie et al., 1999; Bret-Harte et al., 2001; Shaver et al., 2001). In NUCOM-tundra, competition for light or nutrients is determined by a combination of several plant traits (parameters values) . For light competition, leaf area and light extinction are important, which are influenced by parameters as specific leaf area, biomass allocation to leaves, leaf mortality and light extinction coefficient. Similarly, nutrient competition is influenced by parameters as specific root length, root distribution and nutrient requirements. Although tiller production is not explicitly included in NUCOM-tundra, shrubs had an advantage in the competition for light as they have a higher specific leaf area and higher light extinction coefficient than the graminoids. Yet, this advantage for shrubs was in the combined climate change scenarios overruled by the advantageous ability of graminoids to root deep in the mineral soil layers.  Based on our observations at our study site in north-eastern Siberian tundra and at Toolik Lake moist tussock tundra, we assumed the (dwarf) shrubs and graminoids to be equally tall in the model. With warming-induced increases in aboveground biomass for both graminoids and shrubs, the competition for light becomes more important. In parts of the tundra area shrubs, including *B. nana*, can grow taller than the graminoids. For a wider application of the vegetation model, i.e. to include transitions to tall shrub vegetation, a variable plant height up to a maximum would be required.

[revised manuscript text omitted]

a) Temperature and Precipitation change scenario over the 21st century

[revised manuscript text omitted]